# Classical Lagrange Interpolation Based on General Nodal Systems at Perturbed Roots of Unity

**Elías Berriochoa [1],\*,†** , **Alicia Cachafeiro [1],\*,†** , **Alberto Castejón [1],†** and **José Manuel García-Amor [2],†**

[1] Departamento de Matemática Aplicada I, Universidad de Vigo, 36201 Vigo, Pontevedra, Spain; acaste@uvigo.es

[2] Departamento de Matemáticas, Instituto E. S. Valle Inclán, 36001 Pontevedra, Spain; garciaamor@edu.xunta.es

\* Correspondence: esnaola@uvigo.es (E.B.); acachafe@uvigo.es (A.C.);
Tel.: +34-988-387216 (E.B.); +34-986-812138 (A.C.)

† These authors contributed equally to this work.

**Abstract:** The aim of this paper is to study the Lagrange interpolation on the unit circle taking only into account the separation properties of the nodal points. The novelty of this paper is that we do not consider nodal systems connected with orthogonal or paraorthogonal polynomials, which is an interesting approach because in practical applications this connection may not exist. A detailed study of the properties satisfied by the nodal system and the corresponding nodal polynomial is presented. We obtain the relevant results of the convergence related to the process for continuous smooth functions as well as the rate of convergence. Analogous results for interpolation on the bounded interval are deduced and finally some numerical examples are presented.

**Keywords:** lagrange interpolation; unit circle; nodal systems; separation properties; perturbed roots of the unity; convergence

## 1. Introduction

The polynomial interpolation is a classical subject that has been widely studied under different approaches like Lagrange, Hermite, Birkhoff, Pál-type interpolation and some others. Although it is obvious that the subject is important by itself, its numerous numerical applications like numerical integration or numerical derivation are not less important and indeed the polynomial interpolation continues being a subject of current research.

Lagrange interpolation is a very good tool although it is known that for this interpolation scheme and for good nodal systems such as the classical Chebyshev nodes there exists a continuous function on $[-1, 1]$ for which the Lagrange interpolation polynomial diverges (see [1]). A similar problem has been posed for arbitrary arrays and it was proved in [2] that for each nodal array in $[-1, 1]$, there exists a continuous function such that the Lagrange polynomial interpolation diverges almost everywhere. In any case, recalling the words written by Trefethen in his paper [3] we can say that there is nothing wrong with Lagrange polynomial interpolation. "Yet the truth is, polynomial interpolants in Chebyshev points always converge if f is a little bit smooth". As a consequence, to obtain better results one needs to assume better properties on the function to be interpolated like bounded variation or a condition on its modulus of continuity. Thus one of the most important questions in relation with the interpolation of functions is the choice of the interpolation arrays or nodal systems for which one can expect to obtain pointwise or uniform convergence to the function to be interpolated and another important issue is to determine the class of functions for which some type of convergence can be guaranteed. The nodal

systems strongly normal or normal, introduced by Fejér, play an important role in the interpolation theory, although from a practical point of view, the difficulty of testing the definition makes the applications of these systems quite limited. For these systems Grünwald studies in [4] the convergence of the polynomials of Lagrange interpolation for functions satisfying a Lipschitz condition.

Most of the research obtain results on the convergence from the distribution properties of the nodal points. Indeed it was Fejér, who was the first to invert the problem, trying to deduce separation properties of the nodal systems from the interpolation results. The importance of this idea is avaled by the fact that the required interpolation properties are easily verified.

In [5] it is proved that strongly normal distributions on $[-1, 1]$ give quasi uniformly nodal systems on the unit circle, that is the length of the arcs between two consecutive nodes has the order of $\frac{1}{n}$. Although the situation more widely studied corresponds to the bounded interval, there are important results in some other situations in the complex plane like the unit circumference, (see [6]). By taking into account that continuous functions on the unit circle can be approximated by Laurent polynomials, the interpolation polynomials on the unit circumference are constructed in this Laurent space. In this field of research, [7] deserves to be highlighted. There, the nodal points are the roots of complex numbers with modulus 1 and in this case it is obtained a result about convergence of the interpolants for continuous functions with a suitable modulus of continuity. Moreover, in the same paper the problem of extending the results to general nodal systems. Indeed, since the roots of complex numbers with modulus 1 can be interpreted like the zeros of the para-orthogonal polynomials with respect to the Lebesgue measure, now the natural extension is to consider the zeros of the para-orthogonal polynomials with respect to other measures.

In [8] we have generalized the results given in [7] for these new nodal systems. First we work with nodal systems characterized by fulfilling some properties of boundedness, which are suggested by those fulfilled by the roots of unimodular complex numbers, obtaining a result of convergence for continuous functions with a suitable modulus of continuity. Next, by taking into account that the zeros of the para-orthogonal polynomials with respect to measures in the Szegő class (see [9]) with analytic extension up to $|z| > 1$ satisfy the properties that we need, we obtain a similar result about convergence for these type of nodal systems.

In [10] we have studied the Lagrange interpolation process for piecewise continuous functions with suitable properties and by using as nodal points the zeros of the para-orthogonal polynomials with respect to analytic weights, which constitutes a novel approach to the Lagrange interpolation theory.

Another extension to more general nodal systems is given in [11] where it has opened a new trend to interpolation at perturbed roots of unity and the functions to be interpolated belong to the disc algebra.

Now, in the present paper we assume a distribution for the nodes that can be obtained through a perturbation of the uniform distribution and, in particular of the roots of the unity, and which is more general than that given in [11]. Thus in the present paper we start from a different point of view because we base it on properties satisfied by the nodal systems and we do not need to consider orthogonality nor para-orthogonality with respect to any measure on the unit circle. The interpolation arrays are described by a separation property and the main goal is to obtain the properties that play a role in the Lagrange process, as well as to present some relevant examples.

The organization of the paper is the following. In Section 2 we introduce the nodal systems that we use throughout all the paper and we prove the main properties that they satisfy in several propositions. Section 3 is devoted to the Lagrange interpolation problem. First we obtain the Lebesgue constant of the process and then we study the convergence of the Lagrange interpolation polynomials related to continuous functions with appropriate modulus of continuity. Secondly we analyze the rate of convergence when we deal with smooth functions, (see [12]) and we also deduce analogous behavior for interpolation on the bounded interval. The last section is devoted to give some numerical examples.

## 2. Some General Nodal Systems on the Unit Circle

The aim of this paper is to study interpolation problems on the unit circle $\mathbb{T} = \{z \in \mathbb{C} : |z| = 1\}$ by using nodal systems satisfying some suitable properties.

We denote the nodal polynomials by $W_n(z)$ and their zeros by $\{\alpha_{j,n}\}_{j=1}^n$, that is, we assume that

$$W_n(z) = \prod_{j=1}^n (z - \alpha_{j,n}), \text{ where } |\alpha_{j,n}| = 1 \text{ for } j = 1, \cdots, n, \text{ and } \alpha_{j,n} \neq \alpha_{k,n} \text{ for } j \neq k.$$ For simplicity, in the sequel we omit the subscript $n$ and we write $\alpha_j$ instead of $\alpha_{j,n}$ for $j = 1, \cdots, n$. We denote the length of the shortest arc between any two points of the unit circle, $z_1$ and $z_2$, by $z_1 \overasciicircumscript z_2$, and we use the Landau's notation for complex sequences, denoting by $a_n = \mathcal{O}(b_n)$ if $|\frac{a_n}{b_n}|$ is bounded.

Throughout all the paper we assume that the zeros $\{\alpha_j\}_{j=1}^n$ of the nodal polynomials $W_n(z)$ satisfy the following separation property: there exists a positive constant $A$ such that for $n > \dfrac{A}{\pi}$ the length of the shortest arc between two consecutive nodes $\alpha_j$ and $\alpha_{j+1}$, satisfies:

$$\alpha_j \frown \alpha_{j+1} = \frac{2\pi}{n} + \frac{A(j)}{n^2} \text{ with } |A(j)| \leq A \ \forall j = 1, \cdots, n, \tag{1}$$

where $\alpha_{n+1} = \alpha_1$, that is, $\alpha_j \frown \alpha_{j+1} = \dfrac{2\pi}{n} + \mathcal{O}(\dfrac{1}{n^2})$.

We use the same $\mathcal{O}$ to denote different sequences. Unless we mention otherwise, the bounds we obtain from (1) will be uniform.

We also consider other nodal polynomials, $\widetilde{W}_n(z)$, well connected with $W_n(z)$. If we denote $\widetilde{W}_n(z) = z^n - \lambda$, with $\lambda = \alpha_1^n$, then $\widetilde{W}_n(z) = \prod_{j=1}^n (z - \beta_j)$, where

$$\beta_j = \sqrt[n]{\lambda}, \ j = 1, \cdots, n, \text{ and } \alpha_1 = \beta_1.$$

Hence it is clear that the separation property satisfied by the zeros $\{\beta_j\}$ of $\widetilde{W}_n(z)$ is

$$\beta_j \frown \beta_{j+1} = \frac{2\pi}{n} \ \forall j = 1, \cdots, n. \tag{2}$$

In this section we obtain in several propositions the main properties of the nodal polynomials $W_n(z)$. First we recall the following well known relations between arcs and strings that we are going to use throughout the whole paper and which is based on the convex character of the arcsin function: If $z_1$ and $z_2$ belong to $\mathbb{T}$ then

$$\frac{2}{\pi}(z_1 \frown z_2) \leq |z_1 - z_2| \leq (z_1 \frown z_2). \tag{3}$$

**Proposition 1.** *If $\{\alpha_j\}_{j=1}^n$ and $\{\beta_j\}_{j=1}^n$, with $\alpha_1 = \beta_1$, are the nodal points satisfying the separation properties (1) and (2) and we assume they are numbered in the clockwise sense, then*

(i)
$$(j-1)(\frac{2\pi}{n} - \frac{A}{n^2}) \leq \alpha_1 \frown \alpha_j \leq (j-1)(\frac{2\pi}{n} + \frac{A}{n^2}), \text{ for } j \geq 1.$$

(ii)
$$(j+1)(\frac{2\pi}{n} - \frac{A}{n^2}) \leq \alpha_{n-j} \frown \alpha_1 \leq (j+1)(\frac{2\pi}{n} + \frac{A}{n^2}), \text{ for } j \geq 0.$$

(iii)
$$\alpha_j \frown \beta_j \leq (j-1)\frac{A}{n^2}, \text{ for } j \geq 1.$$

(iv)
$$\alpha_{n-j} \frown \beta_{n-j} \leq (j+1)\frac{A}{n^2}, \text{ for } j \geq 0.$$

**Proof.** (i) By applying (1) we have $\dfrac{2\pi}{n} - \dfrac{A}{n^2} \leq \widehat{\alpha_1 - \alpha_2} \leq \dfrac{2\pi}{n} + \dfrac{A}{n^2}$ and for $j \geq 3$ it holds $\dfrac{2\pi}{n} - \dfrac{A}{n^2} \leq \widehat{\alpha_{j-1} - \alpha_j} \leq \dfrac{2\pi}{n} + \dfrac{A}{n^2}$. Then if we sum, it is straightforward (i).

(ii) Proceeding in the same way we get $\dfrac{2\pi}{n} - \dfrac{A}{n^2} \leq \widehat{\alpha_n - \alpha_1} \leq \dfrac{2\pi}{n} + \dfrac{A}{n^2}$ and for $j \geq 0$ it holds $\dfrac{2\pi}{n} - \dfrac{A}{n^2} \leq \widehat{\alpha_{n-j-1} - \alpha_{n-j}} \leq \dfrac{2\pi}{n} + \dfrac{A}{n^2}$. Hence, by computing the sums we have (ii).

(iii) We know that $\alpha_1 = \beta_1$ and we distinguish two possibilities depending on the position of $\beta_j$ related to $\alpha_j$. If

$$\widehat{\alpha_1 - \beta_j} + \widehat{\beta_j - \alpha_j} = \widehat{\alpha_1 - \alpha_j},$$

that is,

$$\widehat{\beta_1 - \beta_j} + \widehat{\beta_j - \alpha_j} = \widehat{\alpha_1 - \alpha_j}$$

and we use that $\widehat{\beta_1 - \beta_j} = (j-1)\dfrac{2\pi}{n}$, which is a consequence of (2), and we also take into account (i) we get

$$\widehat{\alpha_j - \beta_j} \leq (j-1)\dfrac{A}{n^2}.$$

The second case corresponds to $\widehat{\alpha_1 - \alpha_j} + \widehat{\alpha_j - \beta_j} = \widehat{\beta_1 - \beta_j}$ and it can be deduced in the same way.

(iv) We proceed like in (iii) distinguishing the following cases $\widehat{\alpha_{n-j} - \alpha_1} = \widehat{\alpha_{n-j} - \beta_{n-j}} + \widehat{\beta_{n-j} - \alpha_1}$ or $\widehat{\beta_{n-j} - \alpha_{n-j}} + \widehat{\alpha_{n-j} - \alpha_1} = \widehat{\beta_{n-j} - \beta_1}$. □

Notice that we can write (iii) and (iv) as follows $\widehat{\alpha_j - \beta_j} = (j-1)\mathcal{O}(\dfrac{1}{n^2})$ and $\widehat{\alpha_{n-j} - \beta_{n-j}} = (j+1)\mathcal{O}(\dfrac{1}{n^2})$.

**Proposition 2.** *Let us assume that the zeros of the nodal polynomials $W_n(z)$ satisfy the separation property (1). Then it holds*

$$|W_n(z)| < 2e^A, \quad \forall z \in \mathbb{T}. \tag{4}$$

*Moreover, it also holds*

$$\dfrac{|W_n'(z)|}{n} < 2e^A \text{ and } \dfrac{|W_n''(z)|}{n^2} < 2e^A, \quad \forall z \in \mathbb{T}.$$

**Proof.** Since $W_n(\alpha_j) = 0$ for every $j$, let us take $z \in \mathbb{T}$, such that $z$ is not a nodal point. In order to obtain the result we renumber the nodes in the clockwise sense in such a way that $\alpha_1$ is the nodal point nearest to $z$. We distinguish two cases depending on whether the node closest to $z$ is turning in the clockwise sense or in the counter clockwise sense from z. If we assume that the situation is given in Figure 1, that is, $\alpha_1$ is turning in the counter clockwise sense from $z$, then we have

$$\widehat{z - \alpha_1} < \dfrac{\widehat{\alpha_1 - \alpha_2}}{2} \leq \dfrac{\pi}{n} + \dfrac{A}{2n^2}.$$

Now we consider the polynomial $\widetilde{W}_n(z) = \prod\limits_{j=1}^{n}(z - \beta_j)$, with $\beta_1 = \alpha_1$ and satisfying (2), introduced at the beginning of the section. Using property (1) we have $\dfrac{\pi}{n} + \dfrac{A}{2n^2} < \dfrac{2\pi}{n} = \widehat{\alpha_1 - \beta_2}$ and then $z - \beta_2 \neq 0$.

Now, for every $j$ it holds $\dfrac{z-\alpha_j}{z-\beta_j}=1+\dfrac{\beta_j-\alpha_j}{z-\beta_j}$ and therefore, by using Proposition 1,

$$\left|\frac{z-\alpha_j}{z-\beta_j}\right|\leq 1+\frac{\pi}{2}\frac{\widehat{\beta_j-\alpha_j}}{\widehat{z-\beta_j}}\leq 1+\frac{\pi}{2}\frac{(j-1)\dfrac{A}{n^2}}{\widehat{z-\beta_j}}.$$

Besides, from property (1) we have $\widehat{z-\beta_2}>\dfrac{\pi}{2n}$ and for $j\geq 3$, $\widehat{z-\beta_j}>\dfrac{\pi}{2n}+\dfrac{(j-2)2\pi}{n}>\dfrac{(j-2)2\pi}{n}$.
Hence $\left|\dfrac{z-\alpha_j}{z-\beta_j}\right|\leq 1+\dfrac{A}{n}$.

Proceeding in the same way and taking into account that for $j\geq 0$ it holds that $\beta_{n-j}\widehat{-\alpha}_{n-j}\leq$
$(j+1)\dfrac{A}{n^2}$ and $\widehat{z-\beta}_{n-j}>(j+1)\dfrac{2\pi}{n}$ we obtain

$$\left|\frac{z-\alpha_{n-j}}{z-\beta_{n-j}}\right|\leq 1+\frac{\pi}{2}\frac{\widehat{\beta_{n-j}-\alpha_{n-j}}}{\widehat{z-\beta}_{n-j}}\leq 1+\frac{\pi}{2}\frac{(j+1)\dfrac{A}{n^2}}{(j+1)\dfrac{2\pi}{n}}=1+\frac{A}{n}.$$

Therefore we have that

$$\frac{|W_n(z)|}{|\widetilde{W}_n(z)|}=\prod_{j=2}^{n}\left|\frac{z-\alpha_j}{z-\beta_j}\right|<e^A\;\forall z,$$

and since $|\widetilde{W}_n(z)|\leq 2$, then we get $|W_n(z)|<2e^A$.

Notice that if the node closest to $z$, $\alpha_1$, is in the clockwise sense from $z$, we can proceed in a similar
way. Indeed $\widehat{z-\alpha_1}<\dfrac{\widehat{\alpha_1-\alpha_n}}{2}\leq\dfrac{\pi}{n}+\dfrac{A}{2n^2}$ and since $\dfrac{\pi}{n}+\dfrac{A}{2n^2}<\widehat{\beta_n-\alpha_1}=\dfrac{2\pi}{n}$ then $z-\beta_n\neq 0$.

The second statement, related to the first and second derivatives of the nodal polynomial, is a
consequence of Bernstein's theorem, (see [13]). □

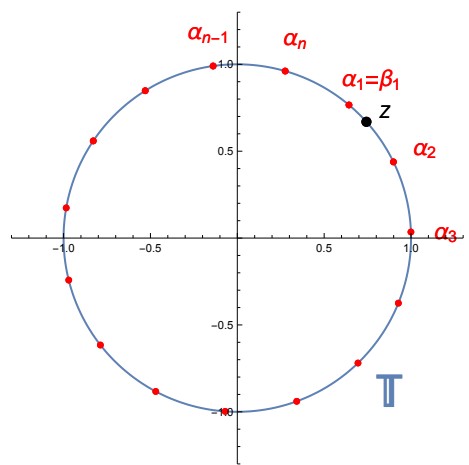

**Figure 1.** An arbitrary point $z$ and the nodal system.

**Proposition 3.** *Let us assume that the zeros of the nodal polynomials $W_n(z)$ satisfy the separation property (1).*
*Then there exists a positive constant $C>0$ such that for $n$ large enough and for every $j=1,\cdots,n$, it holds that*

$$\frac{|W'_n(\alpha_j)|}{n}>C.\tag{5}$$

**Proof.** For simplicity we take $j = 1$ and we try to bound from below $\dfrac{|W_n'(\alpha_1)|}{n}$.

Thus we consider the polynomial $\widetilde{W}_n(z)$ satisfying (2), that is, $\widetilde{W}_n(z) = z^n - \alpha_1^n = \prod\limits_{j=1}^{n}(z - \beta_j)$ with $\alpha_1 = \beta_1$.

Since $\widetilde{W}_n'(z) = nz^{n-1}$ then $|\widetilde{W}_n'(\alpha_1)| = n$ and so our aim is to prove that $\dfrac{|W_n'(\alpha_1)|}{|\widetilde{W}_n'(\alpha_1)|} > C$.

Now, by taking into account that

$$\frac{|W_n'(\alpha_1)|}{|\widetilde{W}_n'(\alpha_1)|} = \prod_{j=2}^{n}\left|\frac{\alpha_1 - \alpha_j}{\alpha_1 - \beta_j}\right|,$$

we study the quotients $\left|\dfrac{\alpha_1 - \alpha_j}{\alpha_1 - \beta_j}\right|$ and $\left|\dfrac{\alpha_1 - \alpha_{n-j}}{\alpha_1 - \beta_{n-j}}\right|$.

On the one hand,

$$\left|\frac{\alpha_1 - \alpha_j}{\alpha_1 - \beta_j}\right| = \left|1 + \frac{\beta_j - \alpha_j}{\alpha_1 - \beta_j}\right| \geq 1 - \left|\frac{\beta_j - \alpha_j}{\alpha_1 - \beta_j}\right|,$$

and by applying (3) and Proposition 1, we have for $j \geq 2$,

$$\left|\frac{\beta_j - \alpha_j}{\alpha_1 - \beta_j}\right| \leq \frac{\pi}{2}\frac{\widehat{\beta_j - \alpha_j}}{\widehat{\alpha_1 - \beta_j}} \leq \frac{\frac{\pi}{2}(j-1)\frac{A}{n^2}}{(j-1)\frac{2\pi}{n}} = \frac{A}{4n},$$

and therefore $\left|\dfrac{\alpha_1 - \alpha_j}{\alpha_1 - \beta_j}\right| \geq 1 - \dfrac{A}{4n}$.

On the other hand,

$$\left|\frac{\alpha_1 - \alpha_{n-j}}{\alpha_1 - \beta_{n-j}}\right| = \left|1 + \frac{\beta_{n-j} - \alpha_{n-j}}{\alpha_1 - \beta_{n-j}}\right| \geq 1 - \left|\frac{\beta_{n-j} - \alpha_{n-j}}{\alpha_1 - \beta_{n-j}}\right|$$

and since for $j \geq 0$ we have

$$\left|\frac{\beta_{n-j} - \alpha_{n-j}}{\alpha_1 - \beta_{n-j}}\right| \leq \frac{\pi}{2}\frac{\widehat{\beta_{n-j} - \alpha_{n-j}}}{\widehat{\alpha_1 - \beta_{n-j}}} \leq \frac{\frac{\pi}{2}(j+1)\frac{A}{n^2}}{(j+1)\frac{2\pi}{n}} = \frac{A}{4n},$$

then $\left|\dfrac{\alpha_1 - \alpha_{n-j}}{\alpha_1 - \beta_{n-j}}\right| \geq 1 - \dfrac{A}{4n}$. Hence

$$\frac{|W_n'(\alpha_1)|}{|\widetilde{W}_n'(\alpha_1)|} = \prod_{j=2}^{n}\left|\frac{\alpha_1 - \alpha_j}{\alpha_1 - \beta_j}\right| \geq (1 - \frac{A}{4n})^{n-1},$$

that is, $|W_n'(\alpha_1)| \geq (1 - \dfrac{A}{4n})^{n-1}n$. Thus, given $\varepsilon > 0$ if $C = e^{-\frac{A}{4}} - \varepsilon > 0$, then for $n$ large enough it holds that $|W_n'(\alpha_1)| > Cn$.

Notice that for another value of $j$ there is no any significant difference. Indeed to obtain that $|W_n'(\alpha_i)| > Cn$, we take the auxiliary polynomial $\widetilde{W}_n(z) = z^n - \alpha_i^n = \prod\limits_{j=1}^{n}(z - \beta_j)$ with $\alpha_i = \beta_i$, we renumber the nodes as in the previous proof and we proceed in a similar way. $\square$

**Proposition 4.** *Let us assume that the zeros of the nodal polynomials $W_n(z)$ satisfy the separation property (1). Then there exists a positive constant $D > 0$ such that*

$$\frac{|W_n(z)|^2}{n^2} \sum_{j=1}^{n} \frac{1}{|z - \alpha_j|^2} < D, \quad \forall z \in \mathbb{T}. \tag{6}$$

**Proof.** Following the same steps of the proof of Lemma 2 in [14] we have

$$|W_n(z)|^2 \sum_{j=1}^{n} \frac{1}{|z - \alpha_j|^2} = |zW_n(z)W_n'(z) + z^2(W_n''(z)W_n(z) - (W_n'(z))^2)|$$

and therefore by using (4) and its consequences in Proposition 2 we get

$$\frac{|W_n(z)|^2}{n^2} \sum_{j=1}^{n} \frac{1}{|z - \alpha_j|^2} \leq \frac{|W_n(z)|}{n} \frac{|W_n'(z)|}{n} + \frac{|W_n''(z)|}{n^2} |W_n(z)| + \frac{|W_n'(z)|^2}{n^2} <$$

$$\frac{B}{n}B + B^2 + B^2, \text{ where } B = 2e^A.$$

□

**Remark 1.** *The nodal systems considered in [15] satisfy condition (1). Indeed they are the para-orthogonal polynomials related to measures in the Baxter class, (see [16]). In that work it is also assumed the additional condition that the sequence $\{(\phi_n^*)'\}$ is uniformly bounded on $\mathbb{T}$, where $\{\phi_n\}$ is the sequence of monic orthogonal polynomials related to the measure and $\{\phi_n^*\}$ is the sequence of the reciprocal polynomials, (see [9]). In that situation studied in [15], properties (4)–(6) also hold. Now, in the present paper we start from a different point of view because we base it on properties satisfied by the nodal systems and we do not need to consider orthogonality nor para-orthogonality with respect to any measure.*

### 3. Lagrange Interpolation. Convergence in Case of Smooth Continuous Functions

To compute the interpolation polynomials, first we recall some well known definitions related to interpolation problems on the unit circle. We work in the space of Laurent polynomials and, in particular, in the subspaces $\Lambda_{p,q}[z] = span\{z^k : p \leq k \leq q\}$, with $p$ and $q$ integers $p \leq q$.

Let us continue denoting by $\{\alpha_j\}_{j=1}^{n}$ the zeros of the the nodal polynomial $W_n(z)$. If $\{u_j\}_{j=1}^{n}$ are arbitrary complex numbers, the Laurent polynomial of Lagrange interpolation $\mathcal{L}_{-E[\frac{n}{2}],E[\frac{n-1}{2}]}(z) \in \Lambda_{-E[\frac{n}{2}],E[\frac{n-1}{2}]}[z]$ characterized by satisfying

$$\mathcal{L}_{-E[\frac{n}{2}],E[\frac{n-1}{2}]}(\alpha_j) = u_j, \text{ for } j = 1, \cdots, n,$$

has the following expression

$$\mathcal{L}_{-E[\frac{n}{2}],E[\frac{n-1}{2}]}(z) = \frac{W_n(z)}{z^{E[\frac{n}{2}]}} \sum_{j=1}^{n} \frac{\alpha_j^{E[\frac{n}{2}]}}{W_n'(\alpha_j)(z - \alpha_j)} u_j.$$

If $F$ is a function and $u_j = F(\alpha_j)$ we denote the corresponding Laurent polynomial $\mathcal{L}_{-E[\frac{n}{2}],E[\frac{n-1}{2}]}(F, z)$. If $n$ is odd, since $E[\frac{n-1}{2}] = E[\frac{n}{2}]$, then the interpolation polynomial $\mathcal{L}_{-E[\frac{n}{2}],E[\frac{n}{2}]}(z) \in \Lambda_{-E[\frac{n}{2}],E[\frac{n}{2}]}[z]$ and when $n$ is even, taking into account that $E[\frac{n-1}{2}] = E[\frac{n}{2}] - 1$, then the Laurent polynomial of Lagrange interpolation $\mathcal{L}_{-E[\frac{n}{2}],E[\frac{n}{2}]-1}(z) \in \Lambda_{-E[\frac{n}{2}],E[\frac{n}{2}]-1}[z]$.

Without loss of generality, to fix ideas and to simplify the notation we assume that the number of nodes is even, $2n$, in which case the interpolation polynomial $\mathcal{L}_{n,n-1}$ belongs to the space $\Lambda_{-n,n-1}$ and it can be written in terms of the fundamental polynomials as follows:

$$\mathcal{L}_{-n,n-1}(z) = \frac{W_{2n}(z)}{z^n} \sum_{j=1}^{2n} \frac{\alpha_j^n}{W_{2n}'(\alpha_j)(z-\alpha_j)} u_j. \tag{7}$$

In order to compute the interpolation polynomials for applications and examples it is more convenient to use the barycentric expression, which is given by

$$\mathcal{L}_{-n,n-1}(z) = \frac{\displaystyle\sum_{j=1}^{2n} \frac{w_j}{z-\alpha_j} u_j}{\displaystyle\sum_{j=1}^{2n} \frac{w_j}{z-\alpha_j}}, \tag{8}$$

with $w_j = \dfrac{\alpha_j^n}{W_{2n}'(\alpha_j)}$, (see [17]).

This last expression has some advantages due to its numerical stability in the sense established in [18]. In this article author claims literally:

*The Lagrange representation of the interpolating polynomial can be rewritten in two more computationally attractive forms: a modified Lagrange form and a barycentric form. We give an error analysis of the evaluation of the interpolating polynomial using these two forms. The modified Lagrange formula is shown to be backward stable. The barycentric formula has a less favourable error analysis, but is forward stable for any set of interpolating points with a small Lebesgue constant. Therefore the barycentric formula can be significantly less accurate than the modified Lagrange formula only for a poor choice of interpolating points.*

So with a good Lebesgue constant (see next Theorem 1) we have good accuracy, at least as good as the intensively used Lagrange interpolation on the Chebyshev nodal systems.

Following [10] we can obtain the Lebesgue constant, (see [19]), and the convergence of this interpolatory process. Notice that this is a novelty result for our general nodal systems satisfying property (1), although the techniques that we use are the same as in [10].

**Theorem 1.** *There exists a positive constant $L > 0$ such that for every function $F$ bounded on $\mathbb{T}$ it holds that*

$$|\mathcal{L}_{-n,n-1}(F,z)| \le L \parallel F \parallel_\infty \log n,$$

*for every $z \in \mathbb{T}$, where $\parallel \parallel_\infty$ denotes the supremum norm on $\mathbb{T}$.*

**Proof.** Let $z$ be an arbitrary point of $\mathbb{T}$ and assume that $z$ is not a nodal point. Then, if we continue assuming the even case, from (7) we get

$$|\mathcal{L}_{-n,n-1}(z)| \le \sum_{j=1}^{2n} \left| \frac{W_{2n}(z)F(\alpha_j)}{W_{2n}'(\alpha_j)(z-\alpha_j)} \right|,$$

and by our hypothesis about $F$ and by Proposition 3 we have

$$|\mathcal{L}_{-n,n-1}(z)| \le \frac{\parallel F \parallel_\infty}{2nC} \sum_{j=1}^{2n} \left| \frac{W_{2n}(z)}{z-\alpha_j} \right|.$$

If we assume that the nodal points closest to $z$ are $\alpha_1$ and $\alpha_{2n}$ then by applying (1) we obtain that for $j > 1$ it holds

$$\widehat{z - \alpha_j} > (j-1)\left(\frac{2\pi}{2n} + \mathcal{O}\left(\frac{1}{4n^2}\right)\right).$$

Thus, by using (3) we obtain

$$\frac{1}{|z-\alpha_j|} < \frac{\pi}{2} \frac{2n}{(j-1)} \frac{1}{(2\pi + \mathcal{O}(\frac{1}{2n}))} = \frac{nE}{j-1},$$

for some positive constant $E$.

Proceeding in the same way we get $z - \widehat{\alpha_{2n-j}} > j(\frac{2\pi}{2n} + \mathcal{O}(\frac{1}{4n^2}))$ and therefore $\frac{1}{|z - \alpha_{2n-j}|} < \frac{nE}{j}$.

We also obtain

$$\left| \frac{W_{2n}(z)}{z - \alpha_1} \right|, \left| \frac{W_{2n}(z)}{z - \alpha_{2n}} \right| < 2nK,$$

for some positive constant $K$.

Indeed $|W_{2n}(z)| = |W_{2n}(z) - W_{2n}(\alpha_1)| = |W_{2n}(e^{i\theta}) - W_{2n}(e^{i\theta_1})| \le 2 \max_{z \in \mathbb{T}} |W'_{2n}(z)||\theta - \theta_1| \le \max_{z \in \mathbb{T}} |W'_{2n}(z)|\pi|z - \alpha_1| \le 2nK|z - \alpha_1|$.

Hence

$$|\mathcal{L}_{-n,n-1}(z)| \le \frac{\| F \|_\infty}{2nC} \left( \left| \frac{W_{2n}(z)}{z - \alpha_1} \right| + \sum_{j=2}^{n} \left| \frac{W_{2n}(z)}{z - \alpha_j} \right| + \sum_{j=1}^{n-1} \left| \frac{W_{2n}(z)}{z - \alpha_{2n-j}} \right| + \left| \frac{W_{2n}(z)}{z - \alpha_{2n}} \right| \right) \le$$

$$\frac{\| F \|_\infty}{2nC} \left( 4nK + 2\sum_{j=2}^{n} \frac{2e^A nE}{j - 1} \right) = \frac{2 \| F \|_\infty}{C} \left( K + \sum_{j=2}^{n} \frac{e^A E}{j - 1} \right) \le 2 \| F \|_\infty P(1 + H_{n-1,1}),$$

with $H_{n-1,1}$ the harmonic number equal to $\sum_{j=1}^{n-1} \frac{1}{j}$ and $P$ a positive constant. $\quad\square$

**Remark 2.** *When the values of $F(\alpha_i)$ are affected by any type of error, which we can suppose is bounded by some $\epsilon > 0$, then the previous result ensures us, taking into account the linearity of the interpolation process, that the final result is affected by an error bounded by $L \log(n) \epsilon$, that is, it is at least so good as the intensively used Lagrange interpolation on the Chebyshev nodal systems.*

For applying the interpolation it could be very useful the following results concerning the convergence and the rate of convergence for smooth continuous functions (see [10,12]).

**Theorem 2.** *(i) Let $F(z)$ be a function defined on $\mathbb{T}$. If $F$ is continuous with modulus of continuity $\omega(F, \delta) = o(|\log \delta|^{-1})$, then $\mathcal{L}_{-n,n-1}(F, z)$ converges uniformly to $F$ on $\mathbb{T}$.*

*(ii) Let $F(z)$ be a function defined on $\mathbb{T}$. If $F(z) = \sum_{-\infty}^{\infty} A_k z^k$ with $|A_k| \le K \frac{1}{|k|^c}$ for $k \ne 0$, with $c > 1$ then $\mathcal{L}_{-n,n-1}(F, z)$ uniformly converges to $F$ on $\mathbb{T}$ and the rate of convergence is $\mathcal{O}\left( \frac{\log n}{n^{c-1}} \right)$.*

*(iii) If $F(z)$ is an analytic function in an open annulus containing $\mathbb{T}$, then $\mathcal{L}_{-n,n-1}(F, z)$ uniformly converges to $F$ on $\mathbb{T}$. Besides, the rate of convergence is geometric.*

**Proof.** The results are consequence of the preceding Theorem 1 and they are also based on the properties satisfied by our nodal systems. Thus one can obtain these results following the same steps as in the proof of Theorems 3 and 4 in [10], where one can see the details. $\quad\square$

*The Case of the Bounded Interval*

We recall that the Lagrange interpolation polynomial $\ell_{n-1}(x)$ related to a nodal system $\{x_j\}_{j=1}^{n}$ in $[-1, 1]$ and satisfying the conditions $\{v_j\}_{j=1}^{n}$ is given by

$$\ell_{n-1}(x) = \sum_{j=1}^{n} \frac{w_n(x)}{w'_n(x_j)(x - x_j)} v_j,$$

where $w_n(x) = \prod_{j=1}^{n} (x - x_j)$. When $v_j = f(x_j)$ for a function $f$, we write $\ell_{n-1}(f, x)$.

In this subsection we consider the nodal polynomial $w_n(x) = \prod\limits_{j=1}^{n}(x - x_j)$ with $\{x_j\}_{j=1}^{n} \subset [-1,1]$ and numbered in the following way: $1 \geq x_1 > x_2 > \cdots > x_{n-1} > x_n \geq -1$.

We also assume that the nodes satisfy the following separation property:

There exists a positive constant $A$ such that for $n > \dfrac{A}{\pi}$

(i) If $x_1 = 1$ and $x_n = -1$ then $\arccos x_{j+1} - \arccos x_j = \dfrac{\pi}{n} + \dfrac{a(j)}{n^2}$, with $|a(j)| \leq A$, $\forall j = 1, \cdots, n-1$.

(ii) If $x_1 < 1$ and $x_n = -1$ then $\arccos x_{j+1} - \arccos x_j = \dfrac{\pi}{n} + \dfrac{a(j)}{n^2}$, with $|a(j)| \leq A$, $\forall j = 1, \cdots, n-1$,

and $2 \arccos x_1 = \dfrac{\pi}{n} + \dfrac{a(0)}{n^2}$, with $|a(0)| \leq A$.

(iii) If $x_1 = 1$ and $x_n > -1$ then $\arccos x_{j+1} - \arccos x_j = \dfrac{\pi}{n} + \dfrac{a(j)}{n^2}$, with $|a(j)| \leq A$, $\forall j = 1, \cdots, n-1$,

and $2(\pi - \arccos x_n) = \dfrac{\pi}{n} + \dfrac{a(n)}{n^2}$, with $|a(n)| \leq A$.

(iv) If $x_1 < 1$ and $x_n > -1$ then $\arccos x_{j+1} - \arccos x_j = \dfrac{\pi}{n} + \dfrac{a(j)}{n^2}$, with $|a(j)| \leq A$, $\forall j = 1, \cdots, n-1$,

and $2 \arccos x_1 = \dfrac{\pi}{n} + \dfrac{a(0)}{n^2}$, with $|a(0)| \leq A$, and $2(\pi - \arccos x_n) = \dfrac{\pi}{n} + \dfrac{a(n)}{n^2}$, with $|a(n)| \leq A$.

Under the above assumptions we obtain the following results about the convergence and the rate of convergence for the interpolation polynomials with these nodal systems.

**Theorem 3.** *If $f$ is a continuous function on $[-1,1]$ and $\omega(f,\delta) = o(|\log\delta|^{-1})$, then the interpolation polynomial $\ell_{n-1}(f,x)$ fulfilling the interpolation conditions*

$$\ell_{n-1}(f, x_j) = f(x_j) \text{ for } j = 1, \cdots, n \tag{9}$$

*converges uniformly to $f$ on $[-1,1]$.*

**Proof.** Let us define a continuous function $F$ on $\mathbb{T}$ by $F(z) = F(\bar{z}) = f(x)$ with $x = \dfrac{z + \dfrac{1}{z}}{2}$. Then it is clear that its modulus of continuity satisfies

$$\omega(F, \delta) = \sup_{z_1, z_2 \in \mathbb{T}, |z_1 - z_2| < \delta} |F(z_1) - F(z_2)| \leq \sup_{x_1, x_2 \in [-1,1], |x_1 - x_2| < \delta} |f(x_1) - f(x_2)| = \omega(f, \delta).$$

To fix ideas we assume that $x_1 \neq 1$ and $x_n \neq -1$, that is, case (iv). By applying Szegő transformation $w_n\left(\dfrac{z + \dfrac{1}{z}}{2}\right) = \dfrac{1}{2^n z^n} \prod\limits_{j=1}^{n}(z - \alpha_j) \prod\limits_{j=1}^{n}(z - \overline{\alpha_j})$, where $\dfrac{\alpha_j + \overline{\alpha_j}}{2} = x_j$, that is, $\alpha_j = e^{i\theta_j}$ with $\theta_j = \arccos x_j$. Hence we consider the nodal polynomial $W_{2n}(z) = 2^n z^n w_n\left(\dfrac{z + \dfrac{1}{z}}{2}\right) = \prod\limits_{j=1}^{n}(z - \alpha_j) \prod\limits_{j=1}^{n}(z - \overline{\alpha_j})$.

Now our nodal system is constituted by $\{\alpha_j\}_{j=1}^{n} \cup \{\overline{\alpha_j}\}_{j=1}^{n}$ and the arguments are $\{\theta_j\}_{j=1}^{n} \cup \{-\theta_j\}_{j=1}^{n}$. If we renumber the arguments in such a way that $-\theta_n = \theta_{n+1}, \cdots, -\theta_1 = \theta_{2n}$, then it holds that

$$\theta_{j+1} - \theta_j = \dfrac{2\pi}{2n} + \dfrac{A(j)}{n^2},$$

with $|A(j)| \leq A$ for $j = 1, \cdots, 2n$ and $\theta_{2n+1} = \theta_1$.

Let $\mathcal{L}_{-n,n-1}(F,z)$ be the Lagrange interpolation polynomial satisfying the conditions

$$\mathcal{L}_{-n,n-1}(F,\alpha_j) = \mathcal{L}_{-n,n-1}(F,\overline{\alpha_j}) = f(x_j), \quad \text{for } j = 1, \cdots, n.$$

Since $F$ satisfies the hypothesis of Theorem 2 (i) then $\mathcal{L}_{-n,n-1}(F,z)$ converges uniformly to $F$ on $\mathbb{T}$.

Analizing the expression of $\mathcal{L}_{-n,n-1}(F,z)$ and by taking into account that $W_{2n}(z)$ as well as $W'_{2n}(z)$ have real coefficients we get that $\mathcal{L}_{-n,n-1}(F,z)$ has real coefficients. Since it is clear that $\mathcal{L}_{-n,n-1}(F,\frac{1}{z}) \in \Lambda_{-(n-1),n}$ satisfies the same interpolation conditions, now we consider the algebraic polynomial $\frac{1}{2}\left(\mathcal{L}_{-n,n-1}(F,z) + \mathcal{L}_{-n,n-1}(F,\frac{1}{z})\right)$, which has real coefficients and satisfies the interpolation conditions (9). Since the polynomial satisfying (9) is uniquely determined, then $\frac{1}{2}\left(\mathcal{L}_{-n,n-1}(F,z) + \mathcal{L}_{-n,n-1}(F,\frac{1}{z})\right) = \ell_{n-1}(f,x)$ and it converges uniformly to $f$ on $[-1,1]$.

When $x_1 = 1$ (case (iii)), or $x_n = -1$ (case (ii)), or $x_1 = 1$ and $x_n = -1$ (case (i)), one proceeds in a similar way and the auxiliary nodal polynomials are given by $W_{2n-1}(z) = \dfrac{2^n z^n}{z-1} w_n\left(\dfrac{z+\frac{1}{z}}{2}\right) =$

$(z-1)\displaystyle\prod_{j=2}^{n}(z-\alpha_j)\prod_{j=2}^{n}(z-\overline{\alpha_j})$ or $W_{2n-1}(z) = \dfrac{2^n z^n}{z+1} w_n\left(\dfrac{z+\frac{1}{z}}{2}\right) = (z+1)\displaystyle\prod_{j=1}^{n-1}(z-\alpha_j)\prod_{j=1}^{n-1}(z-\overline{\alpha_j})$ or

$W_{2n-2}(z) = \dfrac{2^n z^n}{(z-1)(z+1)} w_n\left(\dfrac{z+\frac{1}{z}}{2}\right) = (z-1)(z+1)\displaystyle\prod_{j=2}^{n-1}(z-\alpha_j)\prod_{j=2}^{n-1}(z-\overline{\alpha_j})$, respectively. Notice that in the four cases the nodal polynomials have real coefficients. $\square$

Notice that some well known results related to Lagrange interpolation with Chebyshev and Chebyshev extended nodes are particular cases of the above theorem (see [9,20]).

Notice that from the proof of the above theorem and by applying Theorem 1 we can obtain an analogous bound, the Lebesgue constant, for this interpolatory process, that is, there exists a positive constant $M$ such that

$$|\ell_{n-1}(f,x)| \le M \parallel f \parallel_{\infty} \log n. \tag{10}$$

In order to obtain information concerning the rate of convergence, first we recall the following result about the expansion of an analytic function in a Jacobi series (see [9,21]). For a more actual version see [12].

**Theorem 4.** *Let $f(x)$ be analytic on the closed segment $[-1,1]$. The expansion of $f$ in a Jacobi series, $f(x) \sim \sum_{n=0}^{\infty} a_n P_n^{(\alpha,\beta)}(x)$, is convergent in the interior of the greatest ellipse with foci at $\pm 1$, in which $f$ is regular. The expansion is divergent in the exterior of this ellipse and the sum $R$ of the semi-axes of the ellipse of convergence is $R = \liminf \dfrac{1}{\sqrt[n]{|a_n|}}$.*

Thus, in our conditions we have the following results, which are in concordance with Theorem 2.

**Theorem 5.**     *(i) If $f$ is a function defined on $[-1,1]$ by $f(x) = \sum_{k=0}^{\infty} a_k T_k(x)$, where $T_k(x)$ is the Chebyshev polynomial of degree $k$ and with $|a_k| \le \dfrac{K}{k^s}$, with $k \ne 0$, $K > 0$ and $s > 1$, then the Lagrange interpolation polynomial $\ell_{n-1}(f,x)$ converges to $f$ with rate of convergence $\mathcal{O}\left(\dfrac{\log n}{n^{s-1}}\right)$.*

*(ii) If $f$ is analytic on the closed segment $[-1,1]$, then the Lagrange interpolation polynomial $\ell_{n-1}(f,x)$ converges to $f$ with rate of convergence geometric.*

**Proof.** (i) If we decompose $f$ like $f(x) = f_{1,n-1}(x) + f_{2,n-1}(x) = \sum\limits_{k=0}^{n-1} a_k T_k(x) + \sum\limits_{k=n}^{\infty} a_k T_k(x)$, we have

that $\ell_{n-1}(f_{1,n-1}, x) = f_{1,n-1}(x)$ and $|f_{2,n-1}(x)| \leq \sum\limits_{k=n}^{\infty} |a_k| \leq K \sum\limits_{k=n}^{\infty} \dfrac{1}{k^s} < \dfrac{K}{(s-1)(n-1)^{s-1}}$, where the

last inequality follows from the integral criterion. Thus, by applying the Lebesgue constant obtained

before in (10), we get $|\ell_{n-1}(f_{2,n-1}, x)| \leq M \dfrac{K}{(s-1)(n-1)^{s-1}} \log n$. Hence

$$|f(x) - \ell_{n-1}(f, x)| = |f_{1,n-1}(x) - \ell_{n-1}(f_{1,n-1}, x) + f_{2,n-1}(x) - \ell_{n-1}(f_{2,n-1}, x)| =$$

$$|f_{2,n-1}(x) - \ell_{n-1}(f_{2,n-1}, x)| \leq \dfrac{K}{(s-1)(n-1)^{s-1}} (1 + M \log n) \leq T \dfrac{\log n}{n^{s-1}},$$

for some $T > 0$.

(ii) Since $f$ is analytic on $[-1, 1]$, then it can be analytically continued to a neighborhood of $[-1, 1]$

in the complex plane. Hence the expansion in Chebyshev series $\sum\limits_{n=0}^{\infty} a_n T_n(x)$ converges to $f$ in the

interior of the greatest ellipse with foci at $\pm 1$, known as Bernstein ellipse $E_R$ and the sum $R$ of the

semi-axes of the ellipse of convergence is $R = \liminf \dfrac{1}{\sqrt[n]{|a_n|}}$. Then it holds that $|a_n| \leq Pr^n$, for some

$0 < r < 1$ and $P > 0$.

Proceeding in the same way as before we have $f(x) = f_{1,n-1}(x) + f_{2,n-1}(x) = \sum\limits_{k=0}^{n-1} a_k T_k(x) +$

$\sum\limits_{k=n}^{\infty} a_k T_k(x)$, $\ell_{n-1}(f_{1,n-1}, x) = f_{1,n-1}(x)$ and $|f_{2,n-1}(x)| \leq \sum\limits_{k=n}^{\infty} |a_k| \leq P \dfrac{r^n}{1-r}$. Thus, by applying the

Lebesgue constant we get $|\ell_{n-1}(f_{2,n-1}, x)| \leq MP \dfrac{r^n}{1-r} \log n$ and therefore

$$|f(x) - \ell_{n-1}(f, x)| = |f_{2,n-1}(x) - \ell_{n-1}(f_{2,n-1}, x)| \leq Qr^n \log n \leq Qr_1^n,$$

for some $Q > 0$ and $0 < r < r_1 < 1$. □

## 4. Numerical Examples

We have carried out different numerical experiments to visualize the main contributions of this article. The first examples correspond to the three cases of Theorem 2 and in all of them we work in the following way:

1. We construct the nodal systems in a quite random way. We consider four arcs or sections in the unit circumference $\mathbb{T}$. The first one begins in $\alpha_1 = 1$ and its $\dfrac{n}{4}$ nodes are constructed in counter clockwise sense separated by an angular length $\dfrac{2\pi}{n} + \epsilon$, where the $\epsilon$ are random errors determined by using the uniform distribution in $[\dfrac{A}{n^2} 2\pi, \dfrac{2A}{n^2} 2\pi]$. The fourth section begins in $\alpha_1 = 1$ and its $\dfrac{n}{4}$ nodes are constructed in clockwise sense with arcs of angular length $\dfrac{2\pi}{n} + \epsilon$, where the $\epsilon$ are random errors determined by using the uniform distribution in $[\dfrac{A}{n^2} 2\pi, \dfrac{2A}{n^2} 2\pi]$. The second section begins after the first one and its $\dfrac{n}{4}$ nodes are constructed in counter clockwise sense with arcs of angular length $\dfrac{2\pi}{n} + \epsilon$, where the $\epsilon$ are random errors determined by using the uniform distribution in $[-\dfrac{2A}{n^2} 2\pi, -\dfrac{A}{n^2} 2\pi]$. Finally, in the third section the arcs between the nodes are all equal.

Obviously we use different values of $n$ and we must remark that we obtain always the same results, really we must say similar results because due to our random choice we never have the same nodal system.

2. We consider a test function $F(z)$, that we detail in each example, and we always plot $F(z)$ in black.
3. We consider the Lagrange interpolation polynomial $\mathcal{L}_{-n,n-1}(F,)$, which interpolates the test function at the nodal system. We always plot $\Re(\mathcal{L}_{-n,n-1}(F,))$ in red.

These examples are devoted to visualize the items (i), (ii) and (iii) respectively of Theorem 2.

**Example 1.** *In this example we work with* $F(z) = 1 + 20\left(\dfrac{z+z^{-1}}{2}\right)\sin\left(\dfrac{2}{z+z^{-1}}\right)$ *for* $z \in \mathbb{T}$, *which satisfies the hypotheses of Theorem* 2 *(i). We take* $n = 1000$, $A = 2$ *and we use* (8) *to obtain* $\mathcal{L}_{-n,n-1}(F,)$.

*We represent the function* $F(e^{i\theta})$ *which takes real values and, as we have said, the real part of the interpolation polynomial. Notice that due to its variability, F is a quite difficult function to interpolate. Indeed, it is easy to check that* $F(e^{i\theta})$ *is not differentiable at* $\dfrac{\pi}{2}$.

*We present in Figure* 2 *two graphics. On the left we have a general panoramic of the interpolation along* $\mathbb{T}$ *and we have added the interpolation points in green. We must point up that the interpolatory process is successful where the function has no variability. However, we have an unsuccessful situation where the function has great variability.*

*In the graphic on the right we have a detailed situation between 1.2 and 2, that is near* $\dfrac{\pi}{2}$, *which can help us to understand the problem. According to the theory presented, we must increase the number of nodes to obtain better results in this region.*

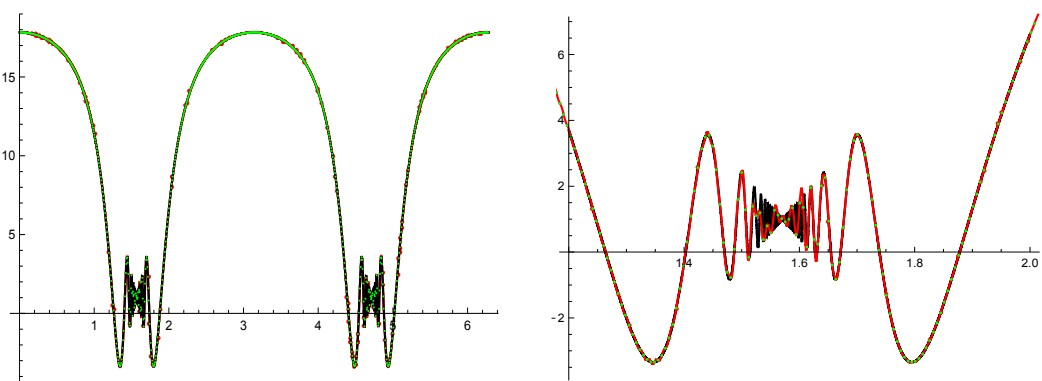

**Figure 2.** $F(z)$ and $\Re(\mathcal{L}_{-n,n-1}(F,z))$ with F(z)= $1 + 20(\dfrac{z+z^{-1}}{2})\sin(\dfrac{2}{z+z^{-1}})$, $z = e^{i\theta}$, $\theta \in [0, 2\pi]$, $\theta \in [1.2, 2]$ and $n = 1000$.

**Example 2.** *Now we consider the function defined on* $\mathbb{T}$ *by* $F(z) = \displaystyle\sum_{k=1}^{\infty}\dfrac{1}{k^6}(z^k + z^{-k})$, *which satisfies the hypotheses of Theorem* 2. *In the next Figure* 3 *we plot on the left* $F(e^{i\theta})$ *and* $\Re(\mathcal{L}_{-n,n-1}(F,e^{i\theta}))$ *for* $\theta \in [0, 2\pi]$ *and* $n = 60$. *Notice that they are indistinguishable. On the right we plot the errors given by* $\Re(\mathcal{L}_{-n,n-1}(F,e^{i\theta})) - F(e^{i\theta})$ *with* $\theta \in [0, 2\pi]$. *We point out that the errors are less or equal than* $2 \times 10^{-8}$.

In the next example we also construct an alternative interpolation polynomial based on the equispaced nodal system on $\mathbb{T}$, but using the values of the function on our nodal system. We do this because a natural criticism to our method could be that with errors as $\mathcal{O}(1/n^2)$ we can be so close to the equispaced nodal system to accept this approximation. We denote by $\mathcal{A}_{-n,n-1}(F,)$ this alternative interpolation polynomial.

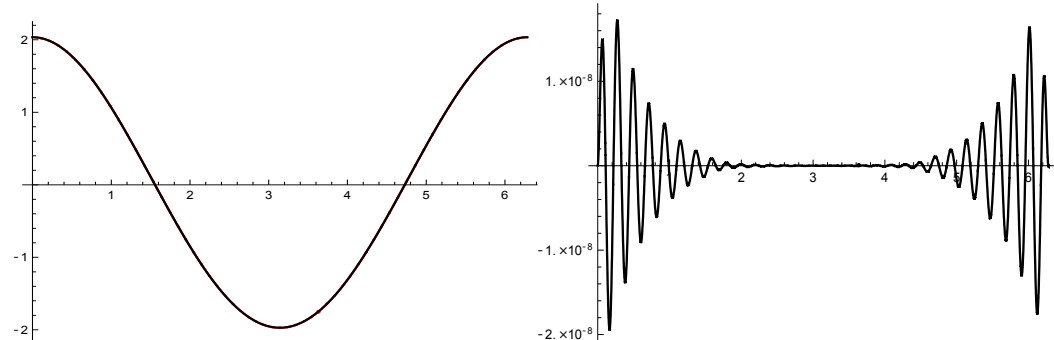

**Figure 3.** $F(z)$ and $\Re(\mathcal{L}_{-n,n-1}(F,z))$ and $\Re(\mathcal{L}_{-n,n-1}(F,z)) - F(z)$ with $F(z) = \sum\limits_{k=1}^{\infty} \frac{1}{k^6}(z^k + z^{-k})$, $z = e^{i\theta}$, $\theta \in [0, 2\pi]$ and $n = 60$.

**Example 3.** *In this example we take $F(z) = e^z$, $n = 24$, $A = 2$ and we use (8) to obtain the interpolation polynomials $\mathcal{L}_{-n,n-1}(F,)$ and $\mathcal{A}_{-n,n-1}(F,)$. Taking into account that $F$ is analytic we know that $F$ and $\mathcal{L}_{-n,n-1}(F,)$ must be close. In Figure 4 we plot $\Re(F)$ in black, $\Re(\mathcal{L}_{-n,n-1}(F,))$ in red and $\Re(\mathcal{A}_{-n,n-1}(F,))$ in green for $z = e^{i\theta}$ with $\theta \in [0, 2\pi]$. On the left hand side we have a global vision with $\theta \in [0, 2\pi]$ and we can observe that $\Re(F)$ and $\Re(\mathcal{L}_{-n,n-1}(F,))$ are indistinguishable; in fact for this example the maximum error was $3 \times 10^{-9}$.*

*Although $\Re(\mathcal{A}_{-n,n-1}(F,))$ has a similar shape, see that it can drive us to catastrophic errors. On the right hand side we present a detail of the previous one, which give us an idea of the error. Notice that in general we cannot know the details of the nodal distribution.*

*We have done the same with the imaginary part and we obtain the same results.*

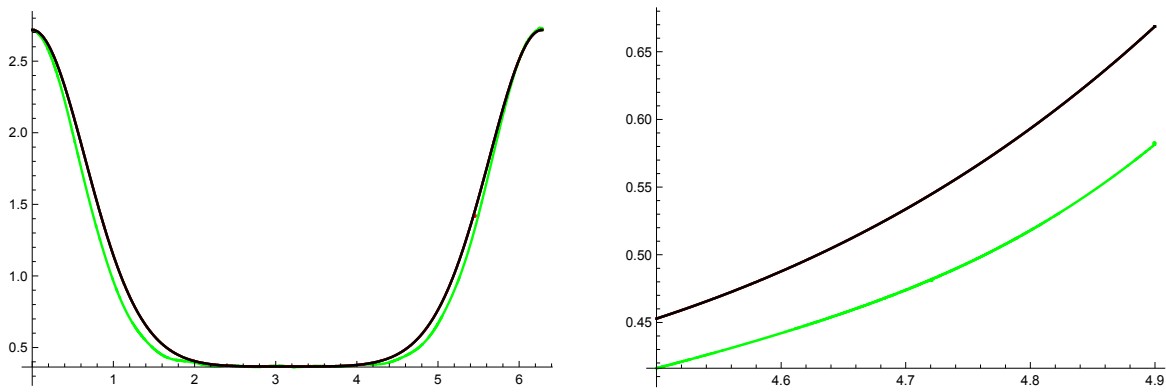

**Figure 4.** $\Re(F(z))$, $\Re(\mathcal{L}_{-n,n-1}(F,z))$ and $\Re(\mathcal{A}_{-n,n-1}(F,z))$ with $F(z) = e^z$, $z = e^{i\theta}$, $\theta \in [0, 2\pi]$, $\theta \in [4.5, 4.9]$ and $n = 24$.

**Example 4.** *Finally we choose $F(z) = \chi_S(z)$ defined on $\mathbb{T}$ as the characteristic function of the superior arc $S$ of $\mathbb{T}$, we take $n = 2000$, $A = 2$ and we use expression (8) to obtain $\mathcal{L}_{-n,n-1}(F,)$. We know the behavior when the nodal system is related to para-orthogonal polynomials with respect to an analytic positive measure (see [10]), but we do not have a theory about the behavior of $\mathcal{L}_{-n,n-1}(F,)$ in our situation. We plot the results in Figure 5. Notice that the basic ideas of the Gibbs–Wilbraham phenomenon are present in this graphic, that is, the convergence of the interpolator to the function in regions which are far enough from the discontinuities and a heavy oscillation near the discontinuities. A representation of the oscillation and its amplitude, maybe, deserves a detailed study.*

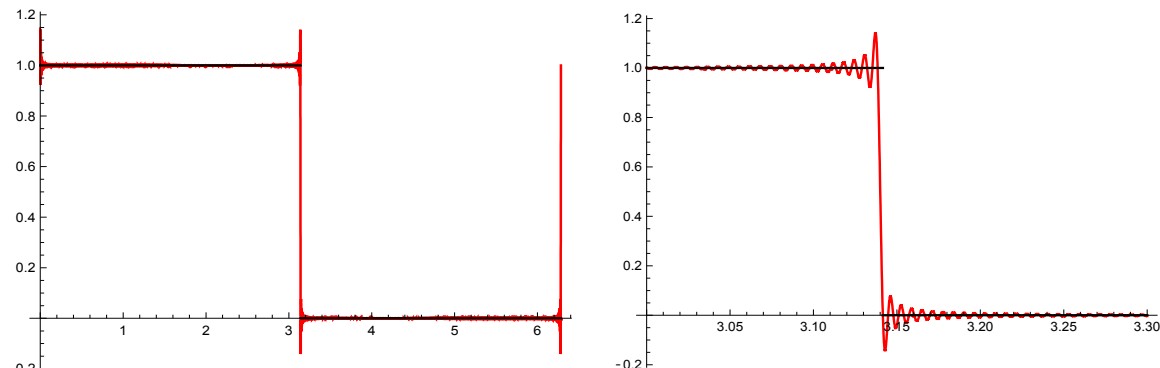

**Figure 5.** $F(e^{i\theta})$ and $\Re(\mathcal{L}_{-n,n-1}(F,e^{i\theta}))$ with $F(z) = \chi_S(z), z = e^{i\theta}, \theta \in [0,2\pi], \theta \in [3,3.3]$ and $n = 2000$.

## 5. Discussion

Usually, the nodal systems used for interpolation problems are strongly connected with measures on the bounded interval and on the unit circle and their corresponding orthogonal or paraorthogonal polynomials. We must point out that these choices are very suitable to construct the whole theory but in some numerical applicatons it is possible that the nodal systems do not satisfy this requisite. So, the starting point of the paper is a distribution for the nodes that can be obtained through a perturbation of the uniform distribution and, in particular of the roots of the unity, and which is more general than that related to measures and orthogonality.

The results of this article contribute to elaborate a theory over these type of nodal systems, as well as to the Lagrange interpolation theory based on these interpolatory arrays. Moreover, a theory about the rate of convergence for some types of smooth functions is given. Finally, we translate the results to perturbed Chevyshev nodal systems and to Lagrange interpolation on the bounded interval.

We think that this research could be of interest for some mechanical models that generate these types of nodal systems. As an example we consider the next problem.

Let us suppose that we are studying a equatorial characteristic $F(e^{i\theta})$ of a planet which depends on the angle $\theta$ and we have a theory which establishes that $F(e^{i\theta})$ is an analytic function. We observe the phenomenon using an observatory in the boundary of a spatial station in an elliptic orbit of period $T$ which rotates over itself with period $T_1$ (with $T = n\,T_1$ and $n$ large enough). Moreover, we take our observations when the center of the planet, the observatory and the center of the station are aligned. We can translate the problem thinking that the planet is our Sun, the spatial station is our Earth and the observatory is our city. So the time between observations is the equivalent of a solar day. It is well known that the duration of a solar day is not constant (see https://en.wikipedia.org/wiki/Equation_of_time for a brief introduction about the so called **Equation of time**), in our case have a little oscillation on $T_1$ and our observations are taken on a nodal system which satisfies (1). Notice that in this case we do not have a equispaced distribution nor the support of the theory of Orthogonal Polynomials. Therefore, before this paper we did not know how to use our data to reconstruct $F(e^{i\theta})$ and after this paper we can be confident about the use of Lagrange interpolation.

Some future research directions could be the study of other types of interpolation on the unit circle and on the bounded interval by using these general interpolatory arrays; as well as to study the correspondig Gibbs–Wilbraham phenomena.

## 6. Materials and Methods

The experiments given in the section Numerical examples were obtained by using personal codes elaborated with Mathematica® 12 (Wolfram Research Europe Ltd, Long Hanborough Oxfordshire, United Kingdom). These programs to obtain the nodal points and to compute the interpolation polynomials as well as the plots of the test functions and their interpolators are available at the public

repository https://www.dropbox.com/sh/0cx9chq3jfzov2w/AAA_SvL2i7HlC7ChMGpuG-Ata?dl=0 There one can find the program related to Example 2. To obtain the other examples some minor changes must be done.

**Author Contributions:** Conceptualization, E.B., A.C. (Alicia Cachafeiro), A.C. (Alberto Castejón) and J.M.G.-A.; Investigation, E.B., A.C. (Alicia Cachafeiro), A.C. (Alberto Castejón) and J.M.G.-A.; Software, E.B., A.C. (Alicia Cachafeiro), A.C. (Alberto Castejón) and J.M.G.-A.; Writing—original draft, E.B., A.C. (Alicia Cachafeiro), A.C. (Alberto Castejón) and J.M.G.-A. All authors have read and agreed to the published version of the manuscript.

**Funding:** This research received no external funding.

**Conflicts of Interest:** The authors declare no conflict of interest.

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
