# Peer review of "Classical Lagrange Interpolation Based on General Nodal Systems at Perturbed Roots of Unity"

_mathematics, doi:10.3390/math8040498_

Round 1

Reviewer 1 Report

\par The paper deals with approximation properties of the Lagrange interpolation when the knots are chosen near of the equidistant knots on the unit circle.

By Szeg\"o transform the results can be applied for interpolation on segment $[-1,1]$. The paper gives an important generalization of previous results obtained in great part by the same authors. In the final part there are given several numerical examples. The article is very interesting. It is well written.

\par We recommend the publication with a minor revision. Bellow there are our few remarks.

\begin{itemize}

\item Page 4. In the proof of Proposition 2, the choice of the knots $\beta_j$ and hence the polynomial $\tilde{W}$, depend on $z$. So that the inequality $\left|\frac{W(z)}{\tilde{W}(z)}\right|\le 2e^A,\;\forall z\in\mathbb{R}$ must be written after the choice of $\alpha_1$. In the actual form there appears a confusion, because it can understand that $\beta_j$ are fixed regarding the initial counting of the knots $\alpha_j$.

\item Page 5, line 102: we have $\widehat{z-\beta_j}>\frac{(j-2)2\pi}n+\frac{\pi}{2n}$. The conclusion $\left|\frac{z-\alpha_j}{z-\beta_j}\right|\le 1+\frac {A}{n}$ remains true.

\item Page 7, line 133. We have $span\{z^k:\;p\le k\le q\}$.

\item Page 11, Theorem 5. It is not written that $T_k(x)$ denote the Chebyshev polynomials.

\end{itemize}

Reviewer 2 Report

The authors proposed the Lagrange interpolation on the unit circle taking only into account the separation properties of the nodal points. A detailed study of the properties satisfied by the nodal system and the corresponding nodal polynomial was presented. They showed the relevant results of the convergence related to the process for continuous smooth functions as well as the rate of convergence. However, some drawbacks are still not be addressed:

(1) The authors have to redraw those figures because of their blurry phenomena especially the scales.

(2) The authors have to compare their results with some numerical results in order to prove their results in the manuscript.

(3) The reviewer interests in the stability of the proposed method when the input data are contaminated by random noise for different issues. Nevertheless, the results do not show in the manuscript.

(4) The content of this manuscript was not enough to be published in this high quality journal.

(5) The reviewer concerns about the error analysis of the proposed scheme in the manuscript.

The reviewer strongly suggests that before accepting the manuscript, the authors require to clarify the above-mentioned points and replies the modified manuscript to the reviewer.

Round 2

Reviewer 2 Report

This revision can be accepted.